# Hybrid Models for Natural Language Reasoning: The Case of Syllogistic Logic

## Abstract

Despite the remarkable progress in neural models, their ability to generalize—a cornerstone for applications like logical reasoning—remains a critical challenge. We delineate two fundamental aspects of this ability: compositionality, the capacity to abstract atomic logical rules underlying complex inferences, and recursiveness, the aptitude to build intricate representations through iterative application of inference rules. In the literature, these two aspects are often confounded together under the umbrella term of generalization. To sharpen this distinction, we investigated the logical generalization capabilities of pre-trained large language models (LLMs) using the syllogistic fragment as a benchmark for natural language reasoning. Though simple, this fragment provides a foundational yet expressive subset of formal logic that supports controlled evaluation of essential reasoning abilities. Our findings reveal a significant disparity: while LLMs demonstrate reasonable proficiency in recursiveness, they struggle with compositionality. To overcome these limitations and establish a reliable logical prover, we propose a hybrid architecture integrating symbolic reasoning with neural computation. This synergistic interaction enables robust and efficient inference—neural components accelerate processing, while symbolic reasoning ensures completeness. Our experiments show that high efficiency is preserved even with relatively small neural components. As part of our proposed methodology, this analysis gives a rationale and highlights the potential of hybrid models to effectively address key generalization barriers in neural reasoning systems.

## 1 Introduction

Neural models have achieved substantial advancements at an accelerated pace in recent years. However, they continue to face challenges in generalizing—a capability that is crucial for tasks such as logical deduction. While they excel at pattern recognition, these models often struggle with the systematicity and robustness required for sound reasoning (Marcus (2018); Lake et al. (2017); Huang & Chang (2023); Mondorf & Plank (2024)) This is particularly evident in their limited capacity to generalize beyond the training data, especially in tasks that demand a deep understanding of compositional structures (Hupkes et al. (2023)).

In this work, we focus on two fundamental and complementary aspects of generalization in the context of logical reasoning: *compositionality* and *recursiveness*. Compositionality (Janssen & Partee (1997)) refers to the principle that the meaning of a complex expression is determined by the meaning of its parts and the rules used to combine them —the ability to abstract from complex structures to process simpler ones. Recursiveness, on the other hand, is the capacity to construct complex representations through the iterative application of a finite set of rules—or the ability to build intricate representations through iterative composition of simpler elements. A simple syllogistic example illustrates this distinction. Consider the inference: "if all $a$ are $b$ and all $b$ are $c$, then all $a$ are $c$." A system demonstrating recursiveness can extend this to "if all $a$ are $b$, all $b$ are $c$, and all $c$ are $d$, then all $a$ are $d$." Compositional generalization requires the ability to understand that a simpler inference—such as "if all $a$ are $b$ and all $b$ are $c$, then all $a$ are $c$"— is a valid component of both simple and more complex inferences. A system that is merely recursive might be able to produce longer chains, but a compositional system truly understands the structure of the argument and can reason about its sub-parts. It is possible for a model to be recursive (i.e., process arbitrarily long chains) without being compositional (i.e., without understanding the meaning of the individual links

in the chain and how they combine). This lack of compositionality is a key limitation of current neural models (Vargas Guzmán et al. (2024), Lake & Baroni (2023))

To investigate this issue, we examine the logical generalization capabilities of large pre-trained language models (LLMs) using syllogistic logic—a well-defined, yet non-trivial, fragment of natural language that captures a fundamental aspect of human reasoning. Syllogistic logic was chosen as a clearly tractable baseline for compositional and recursive generalization, as logics that are too expressive—such as full first-order logic—are computationally intractable. We fine-tuned LLMs on two distinct reasoning tasks: (1) selecting a subset of premises to construct direct proofs, and (2) generating formulas that yield a contradiction, enabling indirect (*reductio ad absurdum*) proofs. These tasks were designed to probe different facets of logical generalization, with premise selection requiring an understanding of the relationships between statements and proof by contradiction testing the ability to reason about counterfactuals and derive logical consequences. To ensure a rigorous evaluation, we trained and tested on multiple knowledge bases generated from controlled synthetic data, which incorporates pseudowords to avoid content bias (Bertolazzi et al. (2024)). Our experiments reveal a significant disparity: while LLMs demonstrate reasonable proficiency in recursive reasoning, they struggle with compositional generalization. Specifically, when trained on simpler inferences, LLMs can recognize analogous simple inferences across different knowledge bases and generalize to a certain extent to more complex inference patterns. However, models trained exclusively on complex inferences exhibit a substantial performance drop when required to identify the underlying simpler components. Moreover, we observed notable differences in performance and generalization across various types of reasoning. This finding highlights a critical gap: current neural models, even large pre-trained ones, fail to generalize reliably across the spectrum of logical reasoning tasks.

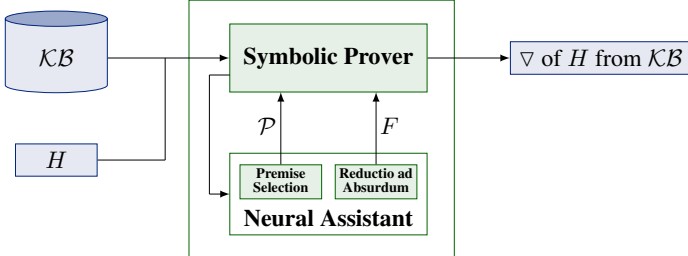

Figure 1: Overview of the hybrid architecture. Input: a knowledge base $\mathcal{KB}$ and a hypothesis $H$. Hybrid Model: the neural models assist the symbolic prover by providing a subset $\mathcal{P} \subset \mathcal{KB}$ such that $\mathcal{P} \vdash H$, and a formula $F$ such that $\mathcal{KB} \cup \{\overline{H}\} \vdash F \wedge \overline{F}$. Output: a proof $\triangledown$ of $H$ from $\mathcal{KB}$.

To address this limitation, we propose a new research program consisting of two elements. On the theoretical side, we aim to understanding how different reasoning building blocks interact with deep-learning model performance on generalization tasks. On the practical side, we develop a novel hybrid architecture (see Figure 1) that integrates the pattern-matching strengths of neural networks with the formal rigor and completeness of symbolic reasoning. In this framework, the neural component serves as an auxiliary to the symbolic prover, efficiently providing candidate premises and formulas to guide the search for proofs. Furthermore, to evaluate the impact of the assistant on the symbolic prover with respect to time complexity (i.e., number of steps), we implemented a relatively straightforward non-deterministic prover. This synergistic approach aims to overcome the limitations of purely neural approaches by enforcing logical consistency and enabling systematic generalization.

The key contributions of this work are: (1) A rigorous empirical demonstration that, despite their recursive capabilities, LLMs lack true compositionality, a crucial requirement for robust logical reasoning. We emphasize the importance of distinguishing between these two properties in evaluations of neural reasoning systems. (2) A hybrid approach that leverages neural networks for efficient inference (e.g., fast premise selection) while relying on symbolic reasoning to guarantee logical completeness and correctness. (3) A novel methodology, embodied in our Hybrid Model (HM), that effectively addresses the generalization barriers in neural reasoning systems, achieving a balance between efficiency and logical soundness. This provides a pathway towards more reliable and trustworthy AI systems.

## 2 A SYLLOGISTIC PROOF SYSTEM

We considered a syllogistic proof system based on Smiley (1973) to implement hybrid models. The formulas "All $a$ are $b$", "No $a$ are $b$", "Some $a$ are $b$", and "Some $a$ are not $b$" correspond to the symbolic representations $Aab$, $Eab$, $Iab$, and $Oab$, respectively. Next, we formally define the syntax and semantics over a language that consists of four *quantifier* symbols $\mathcal{Q} = \{A, E, I, O\}$ and an infinite set of *term* symbols $\mathcal{X} = \{a, b, c, \ldots\}$ denoted by lower-case letters (sometimes with subscripts).

**Syntax and Semantics**   *Well-formed formulas* are built as $Aab$, $Eab$, $Iab$, or $Oab$. An *A-chain*, denoted as $Aa - b$, represents either the formula $Aab$ or the sequence of two or more formulas $Aac_1, Ac_1c_2, \ldots, Ac_{n-1}c_n, Ac_nb$ (for $n \geq 1$). In what follows, when we refer to a formula we mean a well-formed formula. Moreover, we use capital letters (e.g., $F$ or $H$) to denote formulas, and capital calligraphic letters (e.g., $\mathcal{KB}$, $\mathcal{F}$ or $\mathcal{P}$) to denote sets of formulas, unless stated differently.

The meaning of syllogistic formulas can be defined using set-theoretic relationships. Let the terms $a$ and $b$ denote non-empty subsets of an underlying set or universe, then $Aab$ is true *iff* $a \subseteq b$; $Eab$ is true *iff* $a \cap b = \emptyset$; $Iab$ is true *iff* $a \cap b \neq \emptyset$; and $Oab$ is true *iff* $a \not\subseteq b$. A set of syllogistic formulas $\mathcal{F}$ is *consistent* if there exists an interpretation (i.e., an assignment of sets to terms) under which every $F \in \mathcal{F}$ is true.

Note that $Aab$ and $Oab$ are contradictory, and the same about $Iab$ and $Eab$. We denote the negation of a formula $F$ as $\overline{F}$, i.e., $\overline{Aab} = Oab$, $\overline{Oab} = Aab$, $\overline{Iab} = Eab$, and $\overline{Eab} = Iab$. Last but not least, $I$ and $E$-formulas are symmetrical. That is, $Iab$ and $Iba$ have the same meaning, and so do $Eab$ and $Eba$.

**Types of Syllogisms**   We define a syllogism as an inference from a set of premises (or knowledge base) to a conclusion. Unlike classical syllogisms, which typically involve two premises, we consider more complex inferences that may involve multiple premises connected through chains of $A$-formulas.

**Definition 1** (Inference). Let $\mathcal{F}$ be a set of formulas (*premises*) and let $F$ be a formula (*conclusion*). We write $\mathcal{F} \vdash F$, to denote that $F$ is derivable (or provable) from $\mathcal{F}$, if there exists a formal proof of $F$ from $\mathcal{F}$.

**Definition 2** (Proof). The following is a mutually recursive definition to characterize formal proofs for a set of formulas using tree notation. A *proof* $\triangledown$ is one of the following three types:

(i) *Trivial proof*: every $F \in \mathcal{F}$ is a proof from $\mathcal{F}$

$$\frac{}{F}\ ^{(i)}$$

(ii) *Rule-based proofs*: the following four trees are proofs from $\mathcal{F}$. Where $\triangledown'$ and $\triangledown''$ are proofs from $\mathcal{F}$

$$\frac{\overset{\triangledown'}{Aab} \quad \overset{\triangledown''}{Abc}}{Aac}\ ^{(r1)} \qquad \frac{\overset{\triangledown'}{Aab} \quad \overset{\triangledown''}{Ebc}}{Eac}\ ^{(r2)} \qquad \frac{\overset{\triangledown'}{Eba}}{Eab}\ ^{(r3)} \qquad \frac{\overset{\triangledown'}{Aba}}{Iab}\ ^{(r4)}$$

(iii) *Proof by contradiction*: where $\triangledown'$ is a proof from $\mathcal{F} \cup \{\overline{H}\}$ and $\triangledown''$ is a proof from $\mathcal{F}$.

$$\frac{\overset{\triangledown'}{F} \quad \overset{\triangledown''}{\overline{F}}}{H}\ ^{(iii)}$$

**Soundness and Completeness**   The deductive system presented above is based on the framework developed by Smiley (1973). In this system, all formulas that are provable are true in all interpretations (as established in Theorem 4), and conversely, all formulas that are true in all interpretations are derivable within the system (as demonstrated in Theorem 3). It therefore follows that the system is both sound and complete.

**Definition 3** (Minimal Inference). An inference $\mathcal{F} \vdash F$ is *minimal* if for no proper subset $\mathcal{F}' \subsetneq \mathcal{F}$, it is the case that $\mathcal{F}' \vdash F$.

Minimal inferences are essential to design a prover assistant, as they use only the necessary premises to derive a conclusion. Table 1 depicts all types of minimal syllogistic inferences, as outlined in Vargas Guzmán et al. (2024).

Table 1: Types of syllogistic inferences

| Type | Syllogism |
|------|-----------|
| (1) | $\{Aa{-}b, Ac{-}d, Oad\} \vdash Obc$ |
| (2) | $\{Aa{-}b\} \vdash Aab$ |
| (3) | $\{Aa{-}b, Ac{-}d, Aa{-}e, Ede\} \vdash Obc$ |
| (4) | $\{Aa{-}b, Aa{-}c\} \vdash Ibc$ |
| (5) | $\{Aa{-}b, Ac{-}d, Ae{-}f, Iae, Edf\} \vdash Obc$ |
| (6) | $\{Aa{-}b, Ac{-}d, Ebd\} \vdash Eac$ |
| (7) | $\{Aa{-}b, Ac{-}d, Iac\} \vdash Ibd$ |

## 3 SYMBOLIC COMPONENT

We implemented a basic automated syllogistic prover that takes as inputs a knowledge base $\mathcal{KB}$ and a hypothesis $H$. The general process is described in Algorithm 1 and consists of finding a proof (as in Definition 2) for $H$. The prover first tries to derive the hypothesis using proofs of types (i) and (ii) by calling the DERIVE function. If it does not succeed, it tries to prove $H$ by contradiction, i.e., proof of type (iii), using the PBC function. If the hypothesis is valid, the prover returns the derivation steps. If, on the contrary, $H$ is invalid, then the prover will exhaust all possibilities to derive it.

---

**Algorithm 1** Syllogistic Prover

**Input:** A hypothesis $H$ and a knowledge base $\mathcal{KB}$.
**Output:** A proof $\triangledown$ of $H$ from $\mathcal{KB}$ (if $H$ is valid).

1: $\Delta \leftarrow \emptyset$        ▷ set of partial proofs $\Delta \subseteq 2^{\mathcal{F}} \times \mathcal{F} \times \mathsf{ProofType}$
2: **if** DERIVE$(H, \mathcal{KB})$ **or** PBC$(H, \mathcal{KB})$ **then**
3:      $\triangledown \leftarrow$ GET_STEPS$(H, \Delta)$        ▷ get partial proofs that derive $H$
4:      **return** $\triangledown$
5: **else**
6:      **return false**
7: **end if**

---

**Derive function** The central component of the prover is the recursive function DERIVE, described in Algorithm 2. The function takes as input the knowledge base $\mathcal{KB}$ and the hypothesis $H$, and initially checks whether proof of type (i) can be directly applied (line 2). If the base case is not satisfied, the function attempts to derive $H$ (line 5) by non-deterministically searching for a set of formulas $\mathcal{F}$ such that $H$ follows from $\mathcal{F}$ by using rules of inference, i.e., proof of type (ii). This process is applied recursively to each $F \in \mathcal{F}$, continuing until the base case is met or no further applicable rules and formula combinations are available. To prevent redundant computations and potential infinite loops, the algorithm is optimized by storing partial proofs as tuples whenever the base case is reached (lines 3 and 6). These stored derivations are reused if the same inputs are encountered again. Additionally, the system tracks failed derivation attempts to avoid repeating them in subsequent searches. Nevertheless, this search process remains computationally demanding. This is particularly evident when constructing *A-chains*, where the algorithm conducts a non-deterministic search over formulas of the form $Aab$, generated using terms drawn from the knowledge base. In the worst case, this results in at most $n!/(n-2)!$ attempts, where $n$ denotes the number of distinct terms in the knowledge base.

**Proof by Contradiction Function** The final component of the prover, denoted PBC, is specified in Algorithm 3. This function gets the same inputs as DERIVE, and it aims to prove the hypothesis $H$ by contradiction by finding a formula $F$ such that $\mathcal{KB} \cup \{\overline{H}\} \vdash F \wedge \overline{F}$. The algorithm begins by generating all possible contradictory formula pairs $(F, \overline{F})$ that can be constructed using the four quantifiers applied to all terms present in the knowledge base $\mathcal{KB}$. It then iterates through these pairs in search of a contradiction (line 4). If such proof is found, it is stored (line 5); otherwise, the process continues until all pairs have been exhausted (line 9). The search for candidate pairs is performed in a non-deterministic manner, which can result in significant computational cost, as the algorithm calls the DERIVE function for each formula within every potential pair.

---

**Algorithm 2** Derive Recursive Function

---

1: **function** DERIVE($H, \mathcal{KB}$)
2:     **if** $H \in \mathcal{KB}$ **then**                 ▷ base case: all formulas in the knowledge base are derivable
3:         $\Delta \leftarrow \Delta \cup \{(\emptyset, H, (\text{i}))\}$
4:         **return true**
5:     **else if** IS_DERIVABLE($\mathcal{F}, H$) **then**             ▷ find a set $\mathcal{F}$ **s.t.** $\mathcal{F} \vdash_{(\text{ii})} H$
6:         $\Delta \leftarrow \Delta \cup \{(\mathcal{F}, H, (\text{ii}))\}$
7:         **for all** $F \in \mathcal{F}$ **do**
8:             **return** DERIVE($F, \mathcal{KB}$)             ▷ recursive call to the function
9:         **end for**
10:     **else**
11:         **return false**
12:     **end if**
13: **end function**

---

**Algorithm 3** Proof by Contradiction Function

---

1: **initialize:** $\mathcal{V} \leftarrow (\mathcal{Q}, \mathcal{X})$                ▷ vocabulary of quantifiers and terms
2: **function** PBC($H, \mathcal{KB}$)
3:     **for all** $(F, \overline{F}) \in$ ALL_PAIRS($\mathcal{V}$) **do**        ▷ all pairs $(F, \overline{F})$ **s.t.** $F \in \mathcal{Q} \times \mathcal{X} \times \mathcal{X}$
4:         **if** DERIVE($\overline{F}, \mathcal{KB}$) **and** DERIVE($F, \mathcal{KB} \cup \{\overline{H}\}$) **then**
5:             $\Delta \leftarrow \Delta \cup \{(\{F, \overline{F}\}, H, (\text{iii}))\}$
6:             **return true**
7:         **end if**
8:     **end for**
9:     **return false**
10: **end function**

---

## 4   CONNECTIONIST COMPONENT

Our hybrid models integrate two distinct fine-tuned LLM components—one for premise selection and another for identifying contradiction formulas—trained on synthetic data to support the prover.

**Synthetic Data**   A knowledge base $\mathcal{KB}$ can be formally represented as an edge-labeled graph $G = (V, E, \gamma)$, where the set of vertices $V$ are terms from the domain $\mathcal{X}$ and the set of formulas correspond to the set of edges $E \subseteq \{(u, v) \mid u, v \in V \text{ and } u \neq v\}$ along with a labeling function $\gamma : E \rightarrow \mathcal{Q}$ that maps edges to syllogistic quantifiers (see Figure 2 for an example). We produced synthetic knowledge bases by randomly generating graphs such that the resulting knowledge bases are consistent. Furthermore, we imposed the constraint that for every formula $F$ derivable from a given knowledge base $\mathcal{KB}$, there exists a unique subset $\mathcal{P} \subseteq \mathcal{KB}$ such that $\mathcal{P} \vdash F$ is minimal. This property is critical for eliminating redundant derivations of the same hypothesis. We refer to such structures as *non-redundant* knowledge bases (see Appendix A.4.1 for more details).

To convert these structured representations into natural language inputs, terms are replaced with artificially generated pseudowords, and formulas are rendered in textual form (see Figure 3). Models are trained and evaluated using multiple knowledge bases with varied term substitutions and premise permutations, a data augmentation strategy that prevents memorization and improves generalization.

**Fine-Tuning LLMs**   We conducted experiments by fine-tuning two transformer-based architectures, FLAN-T5-base (Raffel et al. (2020)), an *encoder-decoder* model developed by Google AI, and GPT-4o-mini (OpenAI (2024)), a *decoder-only* model developed by OpenAI. Importantly, our goal is not to teach general reasoning through fine-tuning, but to adapt models for specific tasks— premise selection and proof by contradiction—while improvements in reasoning skills may still arise. We therefore view our results as reflecting the overall reasoning capabilities of pre-trained and fine-tuned models. In this setting, for both tasks the input consists of a complete knowledge base paired with a hypothesis to be proven, while the output depends on the task: in premise selection, it is the subset of premises required to derive the hypothesis; in proof by contradiction, it is a formula enabling a type (iii) derivation. In both cases, the models process inputs and produce outputs as plain text sequences.

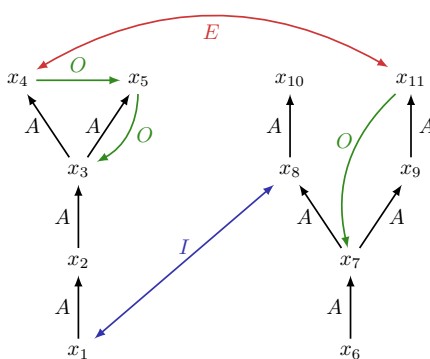

$\mathcal{KB} = \{Ax_1x_2, Ax_2x_3, Ax_3x_4, Ax_3x_5, Ax_6x_7,$
$\quad Ax_7x_8, Ax_7x_9, Ax_8x_{10}, Ax_9x_{11}, Ex_4x_{11},$
$\quad Ix_1x_8, Ox_4x_5, Ox_5x_3, Ox_{11}x_7\}$

Inference examples from each type:
(1) $\{Ax_1\!-\!x_3, Ox_5x_3\} \vdash Ox_5x_1$
(2) $\{Ax_6\!-\!x_{11}\} \vdash Ax_6x_{11}$
(3) $\{Ax_2\!-\!x_4, Ax_7\!-\!x_{11}, Ax_7x_8, Ex_4x_{11}\} \vdash Ox_8x_2$
(4) $\{Ax_7\!-\!x_{10}, Ax_7\!-\!x_{11}\} \vdash Ix_{10}x_{11}$
(5) $\{Ax_1\!-\!x_4, Ax_6\!-\!x_{11}, Ax_8x_{10}, Ex_4x_{11}, Ix_8x_1\} \vdash Ox_{10}x_6$
(6) $\{Ax_1\!-\!x_4, Ax_6\!-\!x_{11}, Ex_4x_{11}\} \vdash Ex_6x_1$
(7) $\{Ax_1\!-\!x_4, Ax_8x_{10}, Ix_8x_1\} \vdash Ix_4x_{10}$

Figure 2: Example of a knowledge base $\mathcal{KB}$ represented as a graph along with valid inferences that can be derived from $\mathcal{KB}$.

We investigate two dimensions of generalization in our framework: *compositionality*, the capacity to deconstruct complex structures into simpler components, and *recursiveness*, the ability to iteratively combine simpler structures to construct more complex ones. We operationalize these concepts by systematically excluding syllogisms with short and long *A-chains* during training and evaluating model performance on them exclusively at test time. More precisely, a model is said to exhibit compositional generalization if it can correctly infer syllogisms with shorter *A-chains* than those encountered during training. Conversely, it demonstrates recursive generalization if it successfully predicts syllogisms with longer *A-chains* than those used in training. To implement this evaluation, we excluded from the training set the five shortest and five longest *A-chain* lengths for each syllogism type.

**Data Specification** To achieve satisfactory performance, we experimented with fine-tuning on varying numbers of knowledge bases and different proportions of the dataset, using 80% of the data for T5 and only 25% for GPT. While OpenAI suggests that small datasets may suffice for fine-tuning, our GPT models were trained on an average of 100 million tokens[1], highlighting the necessity of large-scale data for robust model performance in logical reasoning tasks. We generated 30 distinct synthetic knowledge bases for fine-tuning and an additional 30 for evaluation purposes. On average, each knowledge base consists of 40 premises and 1333 valid hypothesis (see Table 2 for the mean number of hypothesis for each task categorized by syllogism type). To further assess the models' capacity for recursive generalization, we constructed an extra set of 60 knowledge bases, specifically designed to include a greater proportion of inferences involving longer *A-chains*, which are typically underrepresented. Furthermore, each knowledge base features unique pseudoword substitutions and random permutations of premises to enhance lexical diversity and reduce overfitting (see Table 3).

Table 2: Average number of valid hypothesis (by type) for every KB.

| Task | (1) | (2) | (3) | (4) | (5) | (6) | (7) | Total |
|---|---|---|---|---|---|---|---|---|
| Premise Selection | 68 | 152 | 245 | 513 | 42 | 110 | 203 | 1333 |
| Proof By Contradiction | 62 | – | 245 | 361 | 42 | – | 202 | 911 |

Table 3: Dataset specification for fine-tuning LLMs.

| Data split (experiment) | KBs | Substitutions | Permutations |
|---|---|---|---|
| Train (all) | 30 | 10 | 3 |
| Test (overall and compositionality) | 30 | 3 | 1 |
| Test (recursiveness) | 60 | 3 | 1 |

**Generalization Experiments** To assess the generalization capabilities of the fine-tuned models, we compare compositional and recursive variants against *overall* models, i.e., models trained with-

---

[1] Fine-tuning cost: $\approx$ \$3.1k (excluding evaluation and preliminary testing); see Table A13, Appendix A.4.3.

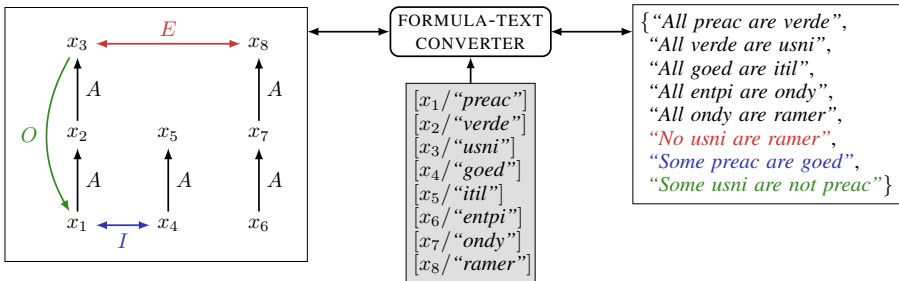

Figure 3: Example of a conversion from a set of syllogistic formulas represented as a graph to a set of formulas in natural language (using pseudoword substitutions) and vice-versa.

out restrictions on *A-chain* lengths. Table 4 presents the average accuracy across all evaluated knowledge bases for each experimental setting, using T5 and GPT architectures fine-tuned on premise selection and proof by contradiction tasks.[2] The evaluation of *overall* models is conducted across three datasets, *all*, *short*, and *long*. The last two are subsets of the first one and correspond to the same tests performed for *compositional* and *recursive* models, respectively.

A more comprehensive analysis is provided in Figure 4, which illustrates the accuracy of evaluated inferences across the five shortest and five longest unseen *A-chain* lengths. For a given syllogism of type $t$, we denote the shortest evaluated length as $\sigma(t)$ and the longest as $\mu(t)$. Solid lines in the plot represent compositional and recursive models, while dashed lines depict overall models. Note that overlapping lines would imply a perfect generalization. The latter is visible in the recursive evaluation for the premise selection task (top-right plot). However, a slight drop in the accuracy occurs as the unseen lengths increase. The drop is more evident for the proof by contradiction task (bottom-right plot). This suggests that LLMs experience difficulties in generating long sequences of $A$-formulas, regardless of their presence during training. Compositional experiments, on the other hand, exhibit a poor generalization and a steeper curve towards the shortest length, in contrast to overall models that can generate inferences involving shorter *A-chains* almost perfectly.

Models occasionally predict unnecessary premises that still lead to valid proofs; when such predictions are treated as correct, GPT exhibits a notable improvement in compositional generalization (see Appendix A.2 for details). Finally, cases in which the models fail to produce the correct answer are analyzed in Appendix A.3.

Table 4: Accuracy scores for premise selection and proof by contradiction tasks (all experiments).

| Experiment | Premise Selection | | Proof By Contradiction | |
|---|---|---|---|---|
| | T5 | GPT | T5 | GPT |
| Overall (all) | $0.94 \pm 0.05$ | $0.94 \pm 0.05$ | $0.93 \pm 0.04$ | $0.95 \pm 0.04$ |
| Overall (short) | $0.99 \pm 0.01$ | $0.98 \pm 0.01$ | $0.99 \pm 0.01$ | $0.98 \pm 0.02$ |
| Overall (long) | $0.79 \pm 0.17$ | $0.83 \pm 0.15$ | $0.76 \pm 0.15$ | $0.88 \pm 0.15$ |
| Compositionality | $0.84 \pm 0.04$ | $0.76 \pm 0.04$ | $0.67 \pm 0.05$ | $0.85 \pm 0.05$ |
| Recursiveness | $0.80 \pm 0.15$ | $0.82 \pm 0.15$ | $0.71 \pm 0.18$ | $0.86 \pm 0.17$ |

**Analysis by Syllogism Type** We analyzed these experiments on specific types of syllogism (Table 1). In the premise selection task, Type (2)—the structurally simplest inference embedded within all other types—consistently achieved the highest accuracy across all experimental settings, demonstrating near-perfect generalization, with the unique exception of GPT in the compositionality experiment. In contrast, Type (1) and Type (5) achieved the lowest accuracies for T5 across the generalization tasks. For GPT, poor performance was observed only on Type (1). In the proof by contradiction task—where Types (2) and (6) were not applicable—Type (7) emerged as the best-performing inference type on both architectures, achieving a near-perfect generalization in the case of GPT. Overall, this task appeared to be easier for GPT than for T5, which, with few exceptions, struggled to generalize effectively. GPT, by contrast, demonstrated consistently strong performance

---

[2]Each experiment was run three times for T5 and twice for GPT; we report the highest accuracy achieved across these runs.

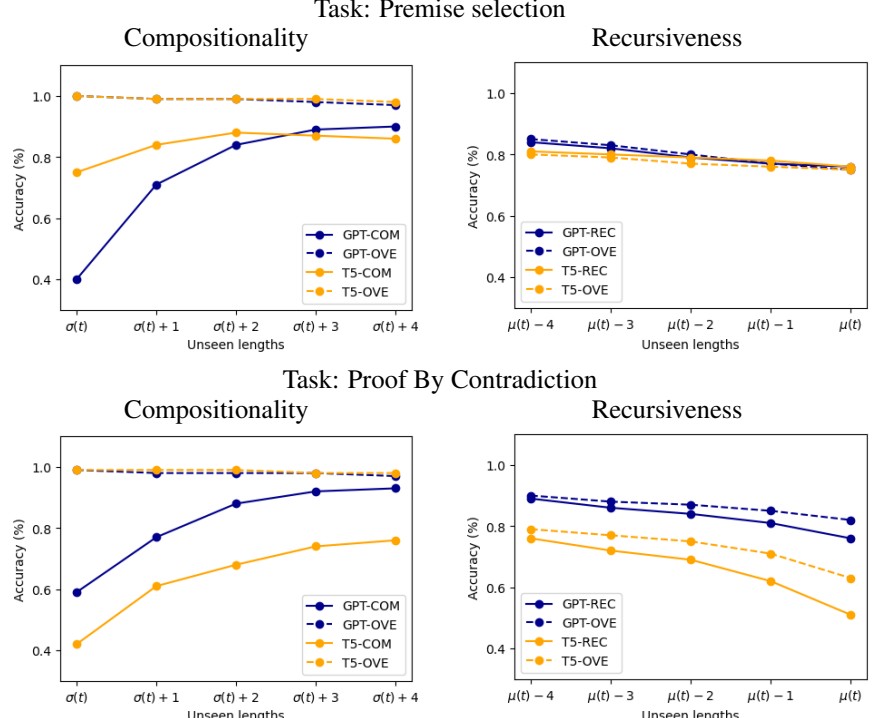

Figure 4: Generalization performance of GPT and T5 architectures across the five shortest (compositional) and five longest (recursive) unseen *A-chain* lengths, denoted as $\sigma(t)$ and $\mu(t)$, respectively, for each syllogism type $t$.

across all inference types, including the particularly challenging Type (1) (see Appendix A.1 for more details).

## 5 HYBRID MODELS EVALUATION

To construct a hybrid model, we employed the same algorithms described in Section 3; however, the search processes are guided by inputs generated by the assistants (neural models). In the case of Algorithm 2, rather than exhaustively exploring all formulas derivable from the knowledge base $\mathcal{KB}$, the hybrid model restricts its search to a subset $\mathcal{P} \subseteq \mathcal{KB}$ which is predicted by the *premise selection* assistant. Similarly, in Algorithm 3 (line 3), the search for contradictory formula pairs $(F, \overline{F})$ is initiated using candidates suggested by the *proof by contradiction* assistant.

We investigated three variants of hybrid models, each incorporating neural components trained under different generalization regimes—namely, overall, compositional, and recursive— and evaluated their performance relative to a purely symbolic baseline to assess the impact of the neural assistants.

**Data Distribution**    We randomly selected just over 2000 samples from 30 distinct knowledge bases within the test dataset. From each knowledge base, we selected 10 syllogisms of each type, comprising 5 instances featuring longer *A-chains* and 5 with shorter *A-chains*. To ensure a sufficient level of inference complexity, we excluded trivial proofs by retaining only those syllogisms in which the *A-chains* length is greater than or equal to 2.

**Evaluation Methodology**    To assess the efficiency of the hybrid models, we measured the number of steps required to complete the proof of a valid hypothesis, where each step corresponds to a single invocation of the recursive function DERIVE by the prover. Due to the non-deterministic nature of the symbolic component, each experiment is executed five times for each model configuration. Moreover, we use logarithmic notation and the geometric mean—due to the potentially wide variability— to represent step counts (see Figure 5).

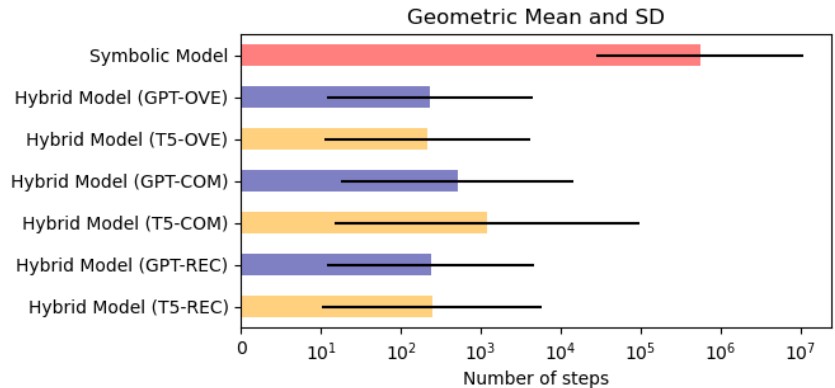

Figure 5: Geometric mean and standard deviation of the number of steps for the Symbolic and Hybrid models, using different assistants trained on GPT and T5. OVE, COM, and REC denote overall, compositional, and recursive models, respectively.

On average, the symbolic model requires approximately $10^{5.7}$ steps to complete a proof. In contrast, hybrid models require substantially fewer steps. In particular, models incorporating overall and recursive neural assistants, exhibit comparable performance, and they need only around $10^{2.4}$ steps across both LLM architectures. This corresponds to a reduction of approximately three orders of magnitude. Hybrid configurations assisted by compositional models require slightly more steps, approximately $10^{2.7}$ for the GPT-based model and $10^{3.1}$ for the T5-based model. This outcome is not unexpected, as they exhibit a drop in accuracy relative to their overall and recursive counterparts. However, their use in hybrid configurations does not result in a significant increase in derivation steps, indicating robustness when assisting the prover.

## 6    CONCLUSIONS

Our main findings focus on the performance of Large Language Models (LLMs) and their role in assisting automated provers. From a semantic perspective, LLMs struggle to fully grasp logical reasoning. Our generalization experiments highlight a significant gap between recursiveness and compositionality, indicating a need for a deeper theoretical understanding of this issue.

Additionally, we observed notable differences in performance and generalization across various types of reasoning. These findings open up a research agenda aimed at understanding how different reasoning building blocks interact with deep learning model performance on generalization tasks. Future research should explore increasingly complex fragments of logic, where the interactions between various inference building blocks and reasoning forms become even more fascinating.

We consider a relatively small encoder–decoder model (T5), chosen for its efficiency and strong overall performance on our tasks, alongside a substantially larger decoder-only model (GPT), selected to evaluate a state-of-the-art LLM (see Appendix A.4.2 for details). GPT shows greater efficiency, converging with less data, though both models achieve comparable performance. This suggests that scaling to larger models alone may not be sufficient to overcome the challenges posed by these reasoning tasks. Nevertheless, our results demonstrate that neither the limitations in generalization nor model size prevent LLMs from effectively assisting symbolic provers. On the contrary, this assistance fosters a collaborative relationship between connectionist and symbolic models. While connectionist models can simplify and expedite tasks, symbolic models can still complete them when necessary, making hybrid models an important area for further investigation.

This study focuses on syllogistic logic—a simple fragment of natural language—laying the groundwork for future work on the theoretical relationship between generalization and logical complexity. Subsequent investigations will explore richer fragments, e.g., Pratt-Hartmann (2004) or suitable fragments of modal logic. Our modular approach to hybrid models could provide practical solutions for developing computationally efficient provers for these logics, forming part of a broader effort to determine where the boundary of tractability lies for neural and neuro-symbolic reasoners.

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

# A  APPENDIX

## A.1  EXPERIMENTS BY TYPES OF SYLLOGISM

In this section, we elaborate on the results presented in Table 4 by providing a breakdown of accuracy scores according to specific syllogism types. The outcomes of the premise selection task are reported in Tables A1 and A2 for the T5 and GPT models, respectively. Similarly, Tables A3 and A4 present the results for the proof by contradiction task.

Table A1: T5 accuracy scores by types of syllogism (premise selection).

| Experiment | (1) | (2) | (3) | (4) | (5) | (6) | (7) |
|---|---|---|---|---|---|---|---|
| Overall (all) | $0.90 \pm 0.10$ | $1.00 \pm 0.01$ | $0.85 \pm 0.13$ | $0.97 \pm 0.03$ | $0.72 \pm 0.28$ | $0.97 \pm 0.08$ | $0.96 \pm 0.08$ |
| Overall (short) | $0.96 \pm 0.05$ | $1.00 \pm 0.00$ | $0.96 \pm 0.05$ | $1.00 \pm 0.01$ | $0.71 \pm 0.36$ | $0.99 \pm 0.03$ | $1.00 \pm 0.01$ |
| Overall (long) | $0.58 \pm 0.38$ | $0.97 \pm 0.09$ | $0.61 \pm 0.37$ | $0.77 \pm 0.21$ | $0.65 \pm 0.41$ | $0.84 \pm 0.33$ | $0.76 \pm 0.31$ |
| Compositionality | $0.35 \pm 0.11$ | $0.99 \pm 0.01$ | $0.88 \pm 0.06$ | $0.87 \pm 0.06$ | $0.49 \pm 0.37$ | $0.90 \pm 0.05$ | $0.77 \pm 0.12$ |
| Recursiveness | $0.59 \pm 0.35$ | $0.96 \pm 0.08$ | $0.66 \pm 0.33$ | $0.74 \pm 0.23$ | $0.69 \pm 0.38$ | $0.88 \pm 0.28$ | $0.80 \pm 0.28$ |

Table A2: GPT accuracy scores by types of syllogism (premise selection).

| Experiment | (1) | (2) | (3) | (4) | (5) | (6) | (7) |
|---|---|---|---|---|---|---|---|
| Overall (all) | $0.84 \pm 0.15$ | $1.00 \pm 0.01$ | $0.87 \pm 0.13$ | $0.97 \pm 0.04$ | $0.84 \pm 0.23$ | $0.99 \pm 0.04$ | $0.96 \pm 0.09$ |
| Overall (short) | $0.91 \pm 0.10$ | $1.00 \pm 0.00$ | $0.98 \pm 0.04$ | $0.99 \pm 0.02$ | $0.93 \pm 0.17$ | $1.00 \pm 0.02$ | $0.99 \pm 0.03$ |
| Overall (long) | $0.54 \pm 0.36$ | $0.98 \pm 0.06$ | $0.67 \pm 0.36$ | $0.81 \pm 0.22$ | $0.70 \pm 0.38$ | $0.94 \pm 0.19$ | $0.86 \pm 0.25$ |
| Compositionality | $0.20 \pm 0.11$ | $0.89 \pm 0.04$ | $0.87 \pm 0.07$ | $0.76 \pm 0.05$ | $0.93 \pm 0.16$ | $0.92 \pm 0.02$ | $0.71 \pm 0.12$ |
| Recursiveness | $0.52 \pm 0.38$ | $0.97 \pm 0.06$ | $0.68 \pm 0.35$ | $0.78 \pm 0.24$ | $0.68 \pm 0.39$ | $0.92 \pm 0.23$ | $0.86 \pm 0.24$ |

Table A3: T5 accuracy scores by types of syllogism (proof by contradiction).

| Experiment | (1) | (3) | (4) | (5) | (7) |
|---|---|---|---|---|---|
| Overall (all) | $0.97 \pm 0.05$ | $0.88 \pm 0.09$ | $0.97 \pm 0.03$ | $0.78 \pm 0.29$ | $0.95 \pm 0.07$ |
| Overall (short) | $0.98 \pm 0.04$ | $0.98 \pm 0.04$ | $1.00 \pm 0.00$ | $0.68 \pm 0.39$ | $0.99 \pm 0.03$ |
| Overall (long) | $0.88 \pm 0.21$ | $0.54 \pm 0.33$ | $0.78 \pm 0.20$ | $0.78 \pm 0.33$ | $0.85 \pm 0.25$ |
| Compositionality | $0.23 \pm 0.12$ | $0.95 \pm 0.05$ | $0.66 \pm 0.06$ | $0.36 \pm 0.36$ | $0.83 \pm 0.12$ |
| Recursiveness | $0.82 \pm 0.27$ | $0.37 \pm 0.33$ | $0.77 \pm 0.22$ | $0.84 \pm 0.31$ | $0.80 \pm 0.29$ |

Table A4: GPT accuracy scores by types of syllogism (proof by contradiction).

| Experiment | (1) | (3) | (4) | (5) | (7) |
|---|---|---|---|---|---|
| Overall (all) | $0.97 \pm 0.06$ | $0.92 \pm 0.12$ | $0.97 \pm 0.04$ | $0.85 \pm 0.30$ | $0.97 \pm 0.07$ |
| Overall (short) | $0.96 \pm 0.07$ | $0.97 \pm 0.08$ | $0.99 \pm 0.02$ | $0.78 \pm 0.37$ | $0.98 \pm 0.06$ |
| Overall (long) | $0.93 \pm 0.20$ | $0.81 \pm 0.30$ | $0.87 \pm 0.19$ | $0.96 \pm 0.16$ | $0.92 \pm 0.19$ |
| Compositionality | $0.90 \pm 0.14$ | $0.97 \pm 0.05$ | $0.78 \pm 0.05$ | $0.73 \pm 0.37$ | $0.98 \pm 0.04$ |
| Recursiveness | $0.90 \pm 0.24$ | $0.74 \pm 0.34$ | $0.81 \pm 0.23$ | $0.93 \pm 0.22$ | $0.95 \pm 0.15$ |

## A.2  ANALYSIS OF UNNECESSARY PREMISES

We evaluated the generalization performance of the models in cases where the predicted set of premises contains both the correct premises and additional, unnecessary ones, while still allowing for a valid proof. In this setting, such predictions are treated as correct, and we report a non-minimal accuracy (NM Acc.) metric for inferences evaluated across the five shortest and five longest unseen *A-chain* lengths. The corresponding results are presented in Tables A5 and A6 for the T5 model, and in Tables A7 and A8 for the GPT model. In the tables, the second column presents the accuracy obtained under the minimal inference criterion (as shown in Figure 4) for comparison. Additionally, we report the average number and standard deviation of unnecessary premises predicted by each model.

Table A5: T5 accuracy scores for shorter unseen lenghts (premise selection)

| Length | Accuracy (minimal) | Non-minimal | |
| --- | --- | --- | --- |
| | | NM Acc. | # Unnec. Prem. |
| $\sigma(t)$ | 0.75 | 0.83 | $5.55 \pm 4.66$ |
| $\sigma(t) + 1$ | 0.84 | 0.86 | $5.77 \pm 4.11$ |
| $\sigma(t) + 2$ | 0.88 | 0.88 | $4.76 \pm 4.10$ |
| $\sigma(t) + 3$ | 0.87 | 0.88 | $4.40 \pm 3.79$ |
| $\sigma(t) + 4$ | 0.86 | 0.86 | $5.22 \pm 4.87$ |
| Total | 0.84 | 0.86 | $5.44 \pm 4.48$ |

Table A6: T5 accuracy scores for longer lenghts (premise selection)

| Length | Accuracy (minimal) | Non-minimal | |
| --- | --- | --- | --- |
| | | NM Acc. | # Unnec. Prem. |
| $\mu(t) - 4$ | 0.81 | 0.81 | $5.21 \pm 3.48$ |
| $\mu(t) - 3$ | 0.80 | 0.81 | $6.86 \pm 3.58$ |
| $\mu(t) - 2$ | 0.79 | 0.79 | $7.46 \pm 3.77$ |
| $\mu(t) - 1$ | 0.78 | 0.78 | $2.75 \pm 1.30$ |
| $\mu(t)$ | 0.76 | 0.76 | $5.67 \pm 1.25$ |
| Total | 0.80 | 0.80 | $6.06 \pm 3.63$ |

Table A7: GPT accuracy scores for shorter unseen lenghts (premise selection)

| Length | Accuracy (minimal) | Non-minimal | |
| --- | --- | --- | --- |
| | | NM Acc. | # Unnec. Prem. |
| $\sigma(t)$ | 0.40 | 0.66 | $5.51 \pm 3.53$ |
| $\sigma(t) + 1$ | 0.71 | 0.81 | $5.33 \pm 3.86$ |
| $\sigma(t) + 2$ | 0.84 | 0.88 | $5.28 \pm 3.63$ |
| $\sigma(t) + 3$ | 0.89 | 0.91 | $5.31 \pm 3.78$ |
| $\sigma(t) + 4$ | 0.90 | 0.92 | $5.30 \pm 3.94$ |
| Total | 0.76 | 0.84 | $5.42 \pm 3.66$ |

Table A8: GPT accuracy scores for longer unseen lenghts (premise selection)

| Length | Accuracy (minimal) | Non-minimal | |
| --- | --- | --- | --- |
| | | NM Acc. | # Unnec. Prem. |
| $\mu(t) - 4$ | 0.84 | 0.86 | $4.22 \pm 4.22$ |
| $\mu(t) - 3$ | 0.82 | 0.84 | $4.21 \pm 4.16$ |
| $\mu(t) - 2$ | 0.79 | 0.81 | $3.48 \pm 3.47$ |
| $\mu(t) - 1$ | 0.77 | 0.79 | $3.46 \pm 3.69$ |
| $\mu(t)$ | 0.76 | 0.77 | $3.86 \pm 4.69$ |
| Total | 0.82 | 0.84 | $3.98 \pm 4.04$ |

## A.3 ERROR ANALYSIS

We examined incorrect model predictions that did not involve the inclusion of unnecessary premises, in order to better understand the inferential patterns being captured. Notably, in all such cases, the models consistently generated well-formed formulas. Therefore, the observed errors can be attributed to the incorrect selection or construction of syllogistic formulas, rather than to syntactic malformation.

To evaluate semantic errors in the premise selection task, we considered three key aspects: (1) whether the terms appearing in the hypothesis were also present in the predicted premises, which would indicate a basic understanding of syllogistic rules; (2) whether the predicted premises were sourced from the knowledge base; and (3) whether the terms within the predicted premises were

Table A9: Semantic validity for the premise selection task

| Model | Experiment | Term overlap (with $H$) | Premise validity | Term validity |
|---|---|---|---|---|
| T5 | Overall | 0.77 | 0.35 | 0.92 |
| | Compositionality | 0.46 | 0.25 | 0.90 |
| | Recursiveness | 0.63 | 0.18 | 0.89 |
| GPT | Overall | 0.94 | 0.20 | 0.92 |
| | Compositionality | 0.71 | 0.31 | 0.94 |
| | Recursiveness | 0.92 | 0.32 | 0.92 |

restricted to those found in the knowledge base. These latter checks ensure that models are not generating fabricated content.

Table A9 presents a summary of semantic validity by reporting the proportion of cases, across all experiments, in which each criterion was satisfied for both T5 and GPT models. These proportions were calculated relative to the set of incorrect predictions. Remarkably, the results indicate that the models generate fabricated premises. A closer analysis suggests that this issue may stem from confusion between syllogisms whose hypotheses share the same formula type—specifically, among $O$-formulas in inference types (1), (3), and (5), and among $I$-formulas in types (4) and (7). That is, the models appear to erroneously construct an incorrect type of syllogism, thereby generating fabricated premises in an attempt to force a valid inference.

Similarly, in the proof by contradiction task, all incorrect predictions were syntactically well-formed. While the criteria concerning term overlap with the hypothesis and the validity of premises are not directly applicable in this setting, it is noteworthy that, in all instances, the terms used in the incorrectly predicted formulas were contained within the vocabulary of the knowledge base.

## A.4 EXPERIMENTAL SETUP

All experiments were implemented in Python, with TensorFlow as the primary deep learning framework. The implementation also made use of several additional libraries: Hugging Face's *Transformers* for model loading and fine-tuning; *NumPy* and *Pandas* for numerical operations and data manipulation; *json* and *jsonlines* for data formatting; *os*, *random*, and *itertools* for file handling and data sampling; and *matplotlib* and *pydot* for visualization and graph rendering.

### A.4.1 KNOWLEDGE BASE GENERATION

The process began with the construction of knowledge bases, represented as graph-like structures (see Figure 2). To introduce variability in the data, we focus on two main factors: the number of subgraphs, each corresponding to a tree structure that encodes $A$-formulas, and the maximum length of an $A$-chain within a single subgraph. For both fine-tuning and evaluation, we generated two distinct types of knowledge bases, which were evenly distributed throughout the dataset: those comprising 4 subgraphs, each with a maximum $A$-chain length of 5; and those consisting of 2 subgraphs, with maximum $A$-chain lengths ranging from 7 to 10. Finally, although it is not feasible for a single knowledge base to evenly represent every syllogism type, we ensured that each structure contains a sufficient number of instances for each inference type.

### A.4.2 FINE-TUNING STRATEGY

We conducted preliminary experiments to determine the number of knowledge bases and data proportions that maximize accuracy in the *overall* setting (i.e., when training includes all $A$-chain lengths) across both tasks: premise selection and proof by contradiction. The most effective configuration consisted of 30 graph-based knowledge bases, of which 27 were used for training and 3 for validation. Each base structure was further augmented through substitution with 10 distinct pseudoword sets and presented in 3 different premise orderings. This augmentation process resulted in a total of 900 knowledge bases. A summary of the overall data distribution is provided in Table A10. Finally, we applied a stratified sampling approach to select a proportion of the dataset, based on inference type and $A$-chain length.

Table A10: Data distribution by syllogism type and *A-chain* length (train/validation split)

| Length | (1) | (2) | (3) | (4) | (5) | (6) | (7) | Total |
|---|---|---|---|---|---|---|---|---|
| 0 | 6390 | – | 2400 | – | – | 2400 | 2400 | 13590 |
| 1 | 9060 | 28800 | 4800 | 57600 | 30 | 4800 | 6180 | 111270 |
| 2 | 10170 | 25440 | 8370 | 66720 | 210 | 7140 | 10260 | 128310 |
| 3 | 9510 | 21540 | 11700 | 67920 | 570 | 9060 | 15180 | 135480 |
| 4 | 8430 | 16770 | 15480 | 65700 | 990 | 10620 | 17940 | 135930 |
| 5 | 6300 | 11910 | 18780 | 57120 | 1830 | 11580 | 20220 | 127740 |
| 6 | 4560 | 7530 | 20610 | 44280 | 2580 | 10920 | 19320 | 109800 |
| 7 | 2700 | 4620 | 21420 | 30480 | 3090 | 9960 | 17820 | 90090 |
| 8 | 1380 | 1290 | 20790 | 17940 | 3420 | 8160 | 15420 | 68400 |
| 9 | 570 | 570 | 18900 | 8820 | 3720 | 6420 | 12900 | 51900 |
| 10 | 210 | 150 | 16140 | 4800 | 3930 | 4620 | 8880 | 38730 |
| 11 | 60 | – | 12960 | 1560 | 3420 | 3060 | 2760 | 23820 |
| 12 | 30 | – | 10170 | 540 | 2940 | 2220 | 1380 | 17280 |
| 13 | – | – | 7650 | 120 | 2520 | 1500 | 600 | 12390 |
| 14 | – | – | 5460 | – | 2040 | 900 | 240 | 8640 |
| 15 | – | – | 3570 | – | 1560 | 480 | 60 | 5670 |
| 16 | – | – | 2130 | – | 1140 | 300 | – | 3570 |
| 17 | – | – | 1230 | – | 690 | 120 | – | 2040 |
| 18 | – | – | 510 | – | 300 | – | – | 810 |
| 19 | – | – | 210 | – | 120 | – | – | 330 |
| 20 | – | – | 90 | – | 30 | – | – | 120 |
| Total | 59370 | 118620 | 203370 | 423600 | 35130 | 94260 | 151560 | 1085910 |

Table A11: Data distribution by syllogism type and *A-chain* length (test split for *overall* and *compositional* models)

| Length | (1) | (2) | (3) | (4) | (5) | (6) | (7) | Total |
|---|---|---|---|---|---|---|---|---|
| 1 | 630 | 2847 | 240 | 5694 | 15 | 240 | 240 | 9906 |
| 2 | 879 | 2487 | 480 | 6528 | 48 | 480 | 672 | 11574 |
| 3 | 954 | 2034 | 822 | 6444 | 90 | 678 | 1188 | 12210 |
| 4 | 897 | 1608 | 1176 | 6036 | 162 | 822 | 1656 | 12357 |
| 5 | 753 | 1185 | 1512 | 5202 | 243 | 912 | 1890 | 11697 |
| 6 | 585 | 798 | 1776 | 4116 | 285 | 930 | 1926 | 10416 |
| 7 | 444 | 537 | 1902 | 3030 | 291 | 870 | 1818 | 8892 |
| 8 | 324 | 240 | 1929 | 1908 | 309 | 828 | 1686 | 7224 |
| 9 | 183 | 138 | 1842 | 1146 | 330 | 714 | 1506 | 5859 |
| 10 | 99 | 60 | 1686 | 666 | 318 | 588 | 1194 | 4611 |
| 11 | 60 | – | 1443 | 252 | 327 | 456 | 912 | 3450 |
| 12 | 33 | – | 1215 | 120 | 294 | 348 | 660 | 2670 |
| 13 | 15 | – | 1005 | 42 | 255 | 258 | 372 | 1947 |
| 14 | – | – | 801 | – | 216 | 192 | 198 | 1407 |
| 15 | – | – | 603 | – | 189 | 138 | 84 | 1014 |
| 16 | – | – | 405 | – | 159 | 84 | 42 | 690 |
| 17 | – | – | 264 | – | 120 | 54 | – | 438 |
| 18 | – | – | 150 | – | 93 | 24 | – | 267 |
| 19 | – | – | 78 | – | 57 | – | – | 135 |
| 20 | – | – | 30 | – | 30 | – | – | 60 |
| 21 | – | – | 15 | – | – | – | – | 15 |
| Total | 5856 | 11934 | 19374 | 41184 | 3831 | 8616 | 16044 | 106839 |

Table A12: Data distribution by syllogism type and *A-chain* length (test split for *recursive* models)

| Length | (1) | (2) | (3) | (4) | (5) | (6) | (7) | Total |
|---|---|---|---|---|---|---|---|---|
| $\mu(t) - 4$ | 609 | 2451 | 1107 | 4410 | 549 | 750 | 2958 | 12834 |
| $\mu(t) - 3$ | 408 | 1806 | 684 | 2544 | 444 | 534 | 2196 | 8616 |
| $\mu(t) - 2$ | 237 | 1245 | 348 | 1008 | 306 | 348 | 1446 | 4938 |
| $\mu(t) - 1$ | 111 | 765 | 177 | 444 | 165 | 192 | 756 | 2610 |
| $\mu(t)$ | 42 | 360 | 63 | 168 | 63 | 84 | 288 | 1068 |
| Total | 1407 | 6627 | 2379 | 8574 | 1527 | 1908 | 7644 | 30066 |

For the T5 models, a proportion of 80% was found to be optimal. Fine-tuning was performed for one epoch using the Adam optimizer, with a learning rate of 1e-4 and a batch size of 20. We also explored the effect of different dataset proportions for fine-tuning the GPT models. The results revealed two key findings: first, using only 25% of the data was sufficient to match the performance of the T5 models; second, increasing the proportion to 80% resulted in only marginal performance gains. Therefore, we opted to use the smaller dataset. All GPT models were fine-tuned for one epoch, using a learning rate multiplier of 1.8. Batch sizes, ranging up to 128, were automatically selected by the API based on the dataset size.

The fine-tuning process was performed three times for the T5 model and twice for GPT. In each case, the model achieving the highest accuracy was selected for subsequent experiments with the symbolic prover.

**Pseudoword Handling and OpenAI Moderation Constraints** During the initial stages of model fine-tuning, we encountered challenges related to the use of pseudowords and the OpenAI Moderation API. Although the pseudowords were intentionally meaningless and devoid of semantic content, a significant number of training examples were flagged for violating OpenAI's usage policies—specifically under the category of hate speech—which resulted in the training files being blocked. To resolve this, we introduced delimiters (e.g., "{" and "}") to explicitly mark pseudowords. Interestingly, this restriction applied only during training; the API imposed no such limitations during evaluation.

**Evaluation Settings** We evaluated the fine-tuned models on a set of unseen knowledge bases. To assess the *overall* and *compositional* models, we generated 30 distinct structures. Additionally, to address the underrepresentation of syllogisms involving longer *A-chain* lengths, we constructed an extra set of 60 knowledge bases to include the five longest *A-chain* lengths across all inference types, with the aim of evaluating the *recursive* models. Each structure was instantiated using three distinct sets of pseudowords. Notably, testing across different pseudoword orderings yielded identical results. Therefore, our evaluation excludes permutations of premises. Table A11 presents the data distribution for evaluating the *overall* and *compositional* models, while Table A12 shows the distribution used for the *recursive* models.

### A.4.3 Hardware and Compute Environment

The GPT models were fine-tuned using OpenAI's hosted infrastructure via their fine-tuning API [3]. At the time of experimentation, the cost of fine-tuning the GPT-4o-mini model was $3.00 per one million tokens. Table A13 reports the total number of tokens used during a single run, as well as the corresponding estimated cost for two runs across all experiments and both tasks—namely, premise selection and proof by contradiction.

Experiments involving T5 fine-tuning and hybrid models were conducted on a clustered, Linux-based system using a compute node equipped with an AMD EPYC 7742 64-Core processor, 256 GB of RAM, and a 40 GB NVIDIA A100 GPU.

Table A14 summarizes the total compute time (in hours) used for fine-tuning and evaluating the GPT and T5 models, carried out via the GPT API and the supercomputer, respectively, across all runs, experiments, and tasks.

---

[3](https://platform.openai.com/docs/api-reference/fine-tuning)

Table A13: Resource usage and cost estimation using the GPT API

| Task | Experiment | Tokens (single run) | Total Cost (two runs) |
|---|---|---|---|
| | Overall | 128.40M | $770.37 |
| Premise Selection | Compositionality | 61.14M | $366.84 |
| | Recursiveness | 122.44M | $734.67 |
| | Overall | 80.24M | $481.46 |
| Proof by Contradiction | Compositionality | 42.16M | $252.95 |
| | Recursiveness | 77.25M | $463.52 |

Table A14: Total runtime for all fine-tuning experiments

| Model | Runs | Experiment | Premise Selection | | Proof by Contradiction | |
|---|---|---|---|---|---|---|
| | | | Training Time | Evaluation Time | Training Time | Evaluation Time |
| | | Overall | 16.92 h | 80.00 h | 8.34 h | 8.85 h |
| T5 | 3 | Compositionality | 8.95 h | 56.66 h | 3.96 h | 4.01 h |
| | | Recursiveness | 13.82 h | 82.99 h | 7.94 h | 5.33 h |
| | | Overall | 15.04 h | 11.86 h | 7.53 h | 8.14 h |
| GPT | 2 | Compositionality | 6.38 h | 6.50 h | 4.65 h | 3.76 h |
| | | Recursiveness | 11.82 h | 3.32 h | 7.4 h | 2.26 h |

Finally, Table A15 reports the total runtime (in hours) for the hybrid model evaluation experiments described in Section 5. We precomputed and stored the model outputs to simulate the interactions between neural assistants and the prover in an asynchronous fashion. This approach enables constant-time querying of the neural models, ensuring a more efficient and uniform evaluation. Moreover, real-time requests to the GPT API over the internet were impractical given the scale of the data.

Table A15: Total runtime (5 runs) for the evaluation of hybrid models

| Model | Assistant | Model | Total Time |
|---|---|---|---|
| Symbolic | – | – | 117.14 h |
| | Overall | GPT | 2.58 h |
| | | T5 | 3.15 h |
| Hybrid | Compositional | GPT | 4.27 h |
| | | T5 | 28.92 h |
| | Recursive | GPT | 2.83 h |
| | | T5 | 3.59 h |

