# OpenReview forum: "Hybrid Models for Natural Language Reasoning: The Case of Syllogistic Logic"
_ICLR.cc/2026/Conference — Submitted to ICLR 2026_

### Official Review · Reviewer_me4S · 2025-10-25

**Soundness:** 4
**Presentation:** 3
**Contribution:** 3
**Rating:** 8
**Confidence:** 4

**Summary:**

This paper explores an intriguing question: whether language models can effectively generalize in syllogistic logic reasoning tasks, particularly in terms of compositionality and recursiveness. The study finds that while language models perform reasonably well on recursive generalization, their ability to generalize compositionally remains limited. To overcome these limitations, the authors propose a hybrid model designed to address key generalization barriers in neural reasoning systems. I appreciate the methodology adopted in this paper --- the controlled experiments on language models reflect a rigorous design.

**Strengths:**

1. The paper addresses an intellectually engaging research question.

2. The paper adopts a rigorous and well-structured methodology to explore it.

**Weaknesses:**

1. If the paper could clearly articulate the representative significance of syllogistic logic for reasoning, as well as its potential implications for extending to more complex forms of reasoning, the overall presentation would be further improved.

2. Perhaps pretraining is a more suitable approach than SFT, since it remains uncertain whether existing pretrained models retain knowledge related to syllogistic reasoning [1]. For example, if the pretraining data contains A→B, B→C, A→C, and your constructed dataset contains swa→cdf, cdf→yur, swa→yur, they may not overlap verbatim, but the underlying logical structure is similar. I believe it is important to reflect on whether such structural generalization might influence performance in ways not fully accounted for.

[1] Physics of language models: Part 2.1, grade-school math and the hidden reasoning process. ICLR 2025. https://arxiv.org/abs/2407.20311

**Questions:**

See Weaknesses

---

> ### Author Response · Authors · 2025-11-26
>
> We thank the reviewer for the valuable comments.
>
> 1. We will clarify the representative significance of syllogistic logic in the revised version: syllogisms capture core aspects of deductive reasoning, including logical entailment, premise-conclusion relationships, and compositionality. Historically, they have served as a canonical minimal reasoning framework in logic, philosophy, and cognitive science. By abstracting away from world knowledge content, we can use syllogisms to test structural reasoning rather than memorization. Multi-premise syllogisms introduce longer inference chains, increasing the complexity of the reasoning task. Finally, studying performance on this foundational system could provide insights into how models might generalize to more complex reasoning domains.
>
> 2. If the reviewer refers to the abstract inference pattern *A→B, B→C*, therefore *A→C*, we agree that such a basic form is likely present in pretraining data due to its simplicity and ubiquity. However, our work goes beyond classical two-premise syllogisms by including all syllogistic forms and extending the setting to multi-premise reasoning, for instance, training and evaluating on knowledge bases containing *A-chains* of up to 21 formulas, which is highly uncommon and therefore unlikely to be present in pretrained knowledge. If the concern instead pertains to the structures of the knowledge bases from which models must extract necessary premises, such overlap with pretraining is extremely unlikely: each knowledge base contains approximately 40 premises with fully randomized logical structures and three independent permutations per instance, yielding an astronomically large combinatorial space. Thus, the hybrid models’ performance primarily reflects their ability to learn and generalize new structural inference patterns, rather than rely on memorization or retrieval from pretraining.

---

> ### Comment · Reviewer_me4S · 2025-11-27
>
> I have no further questions. Maybe you can update the manuscript to include some revised claims. Good luck to you.

---

### Official Review · Reviewer_8Nnb · 2025-10-30

**Soundness:** 3
**Presentation:** 3
**Contribution:** 2
**Rating:** 4
**Confidence:** 3

**Summary:**

The paper analyzes the short comings of LLMs in extrapolation performances in the syllogistic reasoning task, and propose to use LLMs as an assistant to a symbolic prover to reduce the number of steps explored.

**Strengths:**

1. The paper performed detailed analysis on a synthetically created syllogistic reasoning task, and exposed the weakness of the current LLMs.
2. The proposed hybrid model can reduce the number of steps need for the solver to find the proof.
3. The paper is written well and easy to follow.

**Weaknesses:**

1. While the paper proposed to have a hybrid model that use the LLMs as assistant to a symbolic solver, it doesn't propose anything to resolve the issue in LLM itself. The paper will also be stronger if the authors can provide analysis on why LLMs cannot extrapolate well in the syllogistic reasoning task.

2.I also want to ask if using LLMs as an assistant to the symbolic solver will actually incur more computational cost than just use the symbolic server alone, given LLM inference can be expensive.

**Questions:**

see weakness.

---

> ### Author Response · Authors · 2025-11-26
>
> We thank the reviewer for the valuable comments.
>
> 1. Our goal in this paper is not to propose a new mechanism that directly resolves the internal limitations of LLMs, but rather to empirically characterize those limitations under controlled conditions and to demonstrate how a principled hybrid architecture can compensate for them in a formal reasoning setting. While we provide an error analysis (see Appendix A.3), understanding *why* neural models fail is an essential step toward developing future solutions.
>
> 2. While we do not provide the time complexity of LLM inference itself, we compare the number of steps required to complete proofs between a purely symbolic model and hybrid architectures using constant-time querying of precomputed neural outputs, which shows approximately a three-order-of-magnitude reduction. The actual evaluation runtimes (Table A15) further illustrate this: the fastest hybrid models complete five runs in around 3 hours, compared to over 100 hours for the symbolic baseline. It is therefore unlikely that the computational cost of LLM inference would surpass that of the symbolic solver. Importantly, we also demonstrate that small models, such as FLAN-T5-Base, are sufficient for the hybrid architecture, requiring modest hardware resources. This highlights one of our key findings: hybrid performance is independent of model scale.

---

### Official Review · Reviewer_1Wfh · 2025-10-31

**Soundness:** 2
**Presentation:** 2
**Contribution:** 1
**Rating:** 2
**Confidence:** 5

**Summary:**

This paper evaluated the generalisation capabilities of Large Language Models (LLMs) in natural language reasoning, focusing specifically on syllogistic reasoning.

This work focused on two important topics in logical reasoning: compositionality and recursiveness and showed that while LLMs perform reasonably well on recursiveness, they exhibit significant difficulty with compositionality.

Authors propose a hybrid architecture that integrates symbolic reasoning with neural computation.

**Strengths:**

Authors use syllogistic logic to explore LLMs' reasoning abilities, and carefully experiment with two fundamental reasoning abilities: composition and recursion.

**Weaknesses:**

To examine whether LLMs can do composition and recursion of syllogism, authors seem to assume that LLMs can do classic syllogistic reasoning. However, recent research already show that LLMs even struggle with single-step syllogistic reasoning.

Tiwalayo Eisape, MH Tessler, Ishita Dasgupta, Fei Sha, Sjoerd van Steenkiste, and Tal Linzen. A
systematic comparison of syllogistic reasoning in humans and language models. In NAACL, 2024.

Andrew K Lampinen, Ishita Dasgupta, Stephanie C Y Chan, Hannah R Sheahan, Antonia Creswell,
Dharshan Kumaran, James L McClelland, and Felix Hill. Language models, like humans, show
content effects on reasoning tasks. PNAS Nexus, 3(7), 2024.

Magdalena Wysocka, Danilo Carvalho, Oskar Wysocki, Marco Valentino, and Andre Freitas. SylloBio-NLI: Evaluating large language models on biomedical syllogistic reasoning. ArXiv:2410.14399, 2025.

Authors propose neurosymbolic approach. This is not new. The update-to-date method is to develop novel extendable neural architecture that can achieve symbolic-level syllogistic reasoning.

Tiansi Dong, Mateja Jamnik, and Pietro Li`o. Neural Reasoning for Sure Through Constructing
Explainable Models. In AAAI, 2025.

**Questions:**

no

---

> ### Author Response · Authors · 2025-11-26
>
> We thank the reviewer for the comments and for highlighting relevant prior work.
>
> 1. The cited studies evaluate pretrained models without task-specific fine-tuning, so their performance primarily reflects knowledge and patterns acquired during pretraining rather than reasoning explicitly learned for the syllogistic task. In our work, we (i) eliminate world-knowledge using pseudowords, (ii) remove structural priors via randomly generated knowledge bases, and (iii) fine-tune models specifically on the reasoning task. This allows us to measure whether models can learn and generalize structural inference patterns, not whether they already possess such patterns from pretraining. Under this controlled and trainable setup, we observe strong overall performance, showing that neural models can acquire reasoning skills when appropriately trained, which justifies subsequently examining their generalization behaviour.
>
> 2. The work of Dong, Jamnik & Liò (AAAI 2025) represents a very promising line of research; however, it follows a fundamentally different paradigm, and we strongly believe that our work remains highly complementary rather than redundant. We explicitly leverage a symbolic prover guided by a neural module, rather than building a neural network that itself performs reasoning. This hybrid design offers distinct trade-offs, including formal proof generation, interpretability through proof steps, and modular integration with existing theorem-proving systems.

---

### Official Review · Reviewer_rK2h · 2025-11-01

**Soundness:** 2
**Presentation:** 3
**Contribution:** 2
**Rating:** 2
**Confidence:** 3

**Summary:**

The paper investigates the logical generalization capabilities of LLMs by distinguishing between two faculties: compositionality and recursiveness. Using syllogistic logic as a controlled benchmark, the authors conclude that LLMs exhibit reasonable recursiveness but struggle with compositionality. To address this limitation, the paper proposes a hybrid neuro-symbolic architecture that integrates neural computation (as a "Neural Assistant") with a formal symbolic prover.

**Strengths:**

1. A key strength is the paper's conceptual distinction between compositionality and recursiveness . This framing helps the research community delve into a more detailed and nuanced analysis of generalization in neural models.

2. The paper is self-contained. While the significance or novelty of the individual components may be debatable, the work clearly identifies a specific problem (poor compositionality) and proposes a complete, functioning solution (the hybrid model) to address it.

3. The use of synthetic data and pseudowords is a methodologically sound practice. It effectively isolates logical form from content bias, which is crucial given that modern LLMs are prone to memorization.

**Weaknesses:**

1. The paper tests FLAN-T5-base and GPT-40-mini. While the results suggest that the compositionality gap is a structural problem, the claim that "scaling to larger models alone may not be sufficient" is not fully proven without testing against today's largest frontier models.

2. The proposed hybrid method, which uses a neural module to reduce the search space for a symbolic prover , is not entirely novel. The core idea is similar to prior work (e.g., Neural Logic Machine) and can be seen as a straightforward application of LLMs to a specific logical domain. Given the paper's core finding that LLMs struggle with compositionality, a more compelling and original contribution would have been a novel method to solve this compositionality problem directly, rather than bypassing it with a hybrid system.

3. I have concerns about the interpretation of the experimental results. The paper defines compositionality as deconstructing complex structures into simpler components and recursiveness as combining simple structures into complex ones. This framing suggests that compositionality is essentially being treated as the reverse process of recursiveness, which is analogous to other known limitations of transformers, such as the "Reversal Curse" [1]. Moreover, recursiveness is an "easy-to-difficult" generalization (training on simple/short chains, testing on complex/long ones) , while compositionality is a "difficult-to-easy" generalization (training on complex/long chains, testing on simple/short ones). Given this setup, the finding that compositionality is more challenging seems straightforward and expected. The authors should provide a more in-depth justification for why this distinction is insightful.


Overall, I have reservations about this paper's core claims, both in its experimental conclusions and its proposed method. I look forward to the authors' response to the weaknesses raised and will reconsider my score based on their answers.

[1] The Reversal Curse: LLMs trained on “A is B” fail to learn “B is A”

**Questions:**

See Weaknesses

---

> ### Author Response · Authors · 2025-11-26
>
> We thank the reviewer for the valuable comments.
>
> 1. We agree that evaluating only two models does not definitively demonstrate whether scaling alone can resolve the compositionality issue. However, comparing two models with a substantial difference in scale and architecture provides meaningful evidence to *suggest* that scaling may not be sufficient, given that both exhibit very similar performance. Moreover, evaluating a broader range of frontier models is beyond the scope of this work and not its primary aim. Our goal is to show that the proposed hybrid method performs robustly across models of markedly different sizes, indicating that its effectiveness is not dependent on model scale.
>
> 2. We acknowledge that the general idea of using neural components to guide symbolic reasoning is not entirely new. However, to the best of our knowledge, there is no prior work that studies syllogistic logic beyond two premises in combination with an automated symbolic reasoner using background knowledge that is assisted by a neural component specifically to support the construction of a formal proof. This setting constitutes our primary research goal and distinguishes our approach from existing frameworks such as Neural Logic Machines, which operate in different logical domains and do not address the same proof-construction task. Furthermore, our contribution is not limited to the architectural design; rather, we offer a systematic and controlled evaluation of hybrid neuro-symbolic reasoning in syllogistic logic, including insights on robustness across model scales and generalization limitations of neural components. We believe this provides important foundational evidence for the design of future neuro-symbolic systems, and complements—rather than bypasses—the investigation of compositionality challenges by demonstrating how hybrid methods can effectively compensate for such limitations.
>
> 3. We agree that, ex post, “difficult-to-easy” generalization might be harder. However, this distinction is not a standard one, and our work is the first to isolate and test it in syllogistic reasoning. Previous studies typically train on short/simple reasoning chains and test on longer ones. Our distinction is insightful for two main reasons:
>
>     - It reveals a non-trivial and previously unmeasured asymmetry in LLM reasoning. If models were learning true rule-like abstractions, shorter proofs—which are strict subparts of longer ones—should not be harder. The fact that LLMs succeed at long-chain extension but fail at extracting and reusing the underlying rule components challenges a widely assumed symmetry in the reasoning literature.
>
>     - It has practical consequences for neuro-symbolic pipelines. Because LLMs excel at recursiveness but struggle with compositionality, symbolic provers benefit unevenly from neural assistance—an effect that cannot be detected without our decomposition.
>
>     Regarding the “Reversal Curse,” the analogy is limited. That work examines generalization under surface-order reversals without differentiating task difficulty or structural complexity. In contrast, our setup offers a principled quantitative comparison of two distinct forms of reasoning generalization, enabling analyses that reversal-based studies cannot capture.

---

### Meta-Review · Area_Chair_VroG · 2026-01-07

**Summary:**

This work delineates two fundamental aspects of this ability, compositionality and recursiveness, by investigating the logical generalization capabilities of pre-trained LLMs using the syllogistic fragment as a benchmark. Experiments show that while LLMs perform reasonably well on recursiveness, they exhibit significant difficulty with compositionality. To overcome these limitations, this work also proposes a hybrid architecture that integrates symbolic reasoning with neural computation.

**Reviewer Concerns:**

During the rebuttal, the authors responded to concerns regarding the significance of syllogistic logic for reasoning, novelty, evaluation analysis on two models, and computational cost. However, concerns remain, as some reviewers note that the main claims about syllogistic logic reasoning under the two proposed aspects, as well as the hybrid architecture integrating symbolic reasoning, are not novel.

**Reviewer Scores:**

The rebuttal clarifications may warrant a slight score increase, but they are not sufficient to change the acceptance decision.

---

### Decision · Program_Chairs · 2026-01-26

Reject